



# Spatial and Temporal Evolution of a Lightning Diagnostic in HWRF (V3.7a)

5    Keren Rosado [1], Bin Liu [2], Vernon Morris [1], Vijay Tallapragada [3], Lin Zhu [2]

[1]Department of Atmospheric Science, Howard University, Washington, DC, 20059, United States of America

[2]NOAA/National Centers for Environmental Prediction/Environment Modeling Center/I. M. Systems Group College Park, 20740, United States of America

[3]NOAA/National Centers for Environmental Prediction/Environment Modeling Center, College Park, 20740, United States of
10   America

*Correspondence to*: Keren Rosado (keren.rosado@noaa.gov)





**Abstract.** The operational Hurricane Weather Research and Forecast (HWRF) model has been used to investigate the role of lightning diagnostics in the life cycle of tropical cyclones. A lightning parameterization, the Lightning Potential Index (LPI), was implemented into HWRF with the motivation that an improvement in the forecast of lightning will lead to reductions in the HWRF model intensity forecast errors and bias. Three questions are addressed: (i) Can the HWRF model predict lightning temporal distributions with an acceptable degree of accuracy? (ii) How well does the HWRF model with lightning parameterization forecast lightning spatial distributions before, during, and after tropical cyclone intensification? (iii) What is the functional relationship between tropical cyclone wind speed and lightning frequency in the HWRF model forecast? A five-day simulation of Idealized tropical cyclones with and without eyewall replacement cycle, has been conducted, followed by two real cases e.g. hurricanes Earl and Igor to evaluate the evolution of the spatial distribution of lightning location. Results from this investigation led to the following observations: (1) the potential for lightning occurrence increases to its maximum peak prior to the maximum predicted wind intensity and (2) the numerical simulations predict a negative correlation between lightning occurrence and maximum winds during the storm's peak intensity.





## 1 Introduction

Tropical cyclones are natural phenomena that affect many countries around the world. Each year these countries have some degree of impact, direct or indirect, to life and property due to this natural phenomenon. Trajectory forecasting of tropical

cyclones has significantly improved during recent years but improving the storm intensity forecast has remained a challenge. This is due to its dependence on a variety of processes across scales and the limited ability of numerical models to represent them, in particular, for the eyewall region. On long time scales, eyewall replacements are difficult to forecast and are one of the most important aspects that affect intensity forecasts. On short time scales, microbursts are a phenomena that are difficult for current operational models to accurately forecast (DeMaria 2007; DeMaria 2014).

With the aim of improving operational tropical cyclones intensity forecasts, we explore the use lightning as a proxy for intensity changes. The formation of lightning in tropical cyclones happens in convective cells where charge separation between super liquid water, graupel, ice, and hail occurs in the presence of strong updrafts. Convection is a critical element in the formation of a tropical cyclone and throughout its lifetime (MacGorman and Rust 1998; Rakov and Uman 2003). Graupel and hail located in the "charging zone" (from 0 °C up to -20 °C) increases reflectivity values associated with strong updrafts. These areas of high

reflectivity have the greatest probability of lightning (MacGorman et al., 1989; MacGorman and Rust 1998).

In the past, several studies have been designed in attempts to establish a relationship between lightning and intensity in tropical cyclones based on observations. Some of these studies are Molinari et al. 1998; Price at al., 2009 and Pan et al., 2010; Pan et al., 2014; DeMaria et al., 2010; DeMaria et al., 2012; and Thomas et al., 2010.

In a study performed by Price et al., (2009), fifty-six (56) hurricanes around the world were analyzed; focusing on establishing a

correlation between lightning occurrence and maximum wind speed in order to investigate whether lightning can be used as a proxy for hurricane intensity changes. They found that in 90% of the cases a lightning peak is observed within 30 hours before the wind speed maximum with an average correlation coefficient of $r = 0.82$ across all cases studied. In another analysis of seven (7) typhoons, by Pan et al., 2010, the lightning peak occurred in the inner core hours before the maximum wind intensity peak. Pan et al., 2014, later expanded the number of samples analyzed to sixty-nine (69) typhoons showing that the lightning peak

occurred before maximum intensity; 78% of the strong typhoons and 56% of the weak typhoons.

Two theories of the relationship between lightning and tropical cyclone intensity change have emerged. One holds that the lightning peak in the outer rain bands is associated with the intensification or rapid intensification of the tropical cyclone (Molinari et al., 1998; Price et al., 2009). Where rapid intensification is defined as an increase in maximum winds speed of 30 *kt* in a period of twenty-four (24) hours (DeMaria et al., 2012). The second theory is that the increase in lightning activity or the

lightning peak in the inner core could be associated with the beginning or end of intensification or rapid intensification period according to Thomas et al., 2010, DeMaria et al., 2012, Abarca et al., (2011) and Stevenson et al., (2014). During the end of the intensification period the updraft velocity in the eyewall increases as well as the graupel concentration within the eyewall (DeMaria et al., 2012; Fierro et al., 2015). These two theories relate the radial location of the lightning occurrences to the effect in storm intensity change.

In most intense tropical cyclones, storm intensity change has been related to eyewall replacement cycles (Kuo et al., 2009). Eyewall replacement cycles are phenomena that involve the emergence of a hurricane eyewall (typically called secondary or



concentric eyewall) radially outside the center eyewall of the (then called primary eyewall) of the storm. Today there is limited understanding of the possible relationship between lightning occurrence and eyewall replacement cycles.

In this study, a diagnostic lightning parameterization scheme called the Lightning Potential Index (LPI) was implemented into the operational Hurricane Weather Research and Forecast (HWRF) model. This parameterization indicates the existence of
lightning based on vertical velocity and hydrometeor mass mixing ratios (Saunders 2008; Yair et al., 2010). The purpose of the implementation is to answer the following three questions:

(i) Can the HWRF model predict lightning temporal distributions with an acceptable degree of accuracy?

(ii) How well does the HWRF with lightning parameterization model forecast lightning spatial distributions before, during, and after tropical cyclone intensification?

(iii) What is the functional relationship between tropical cyclone wind speed and lightning frequency in the HWRF model forecast?

The goal of this investigation is to demonstrate that the relationship observed between tropical cyclone intensity changes and lightning, can be accurately forecasted by a numerical model. In this study, we describe the implementation of the lightning parameterization scheme into the HWRF operational model. First, we tested and evaluated the LPI implementation with an
Idealized frame work (phase 1), second, we tested the LPI implantation with eyewall replacement within an Idealized frame work (phase 2). Third, we tested the LPI implementation with real cases: hurricanes Earl and Igor (phase 3). It was important to first test the LPI implementation with an Idealized frame work because the lightning potential is a diagnostic tool, it cannot change the model forecast of a tropical cyclone.

The paper is organized as follows: In section 2 we describe the lightning parameterization and its implementation into the HWRF
operational model as well as the Idealized frame work. The data for LPI verification and validation is explained as well as real cases synoptic overviews. In section 3 we show results for all the simulations. In Section 4 results are discussed and conclusions of this work were summarized in section 5.

## 2 Methodology

### 2.1 HWRF Model

The HWRF operational model 2015 (V3.7a ) version is an atmospheric – ocean model fully compressible, non-hydrostatic with 61 atmospheric vertical levels up to 2 *hPa* (Tallapragada et al., 2014; 2015). It provides high-resolution real-time tropical cyclone forecast for the Atlantic, Eastern Pacific and Western Pacific Oceans. This model was designed based on the Weather Research and Forecast system (WRF) (Tallapragada et al., 2014; 2015) and the Geophysical Fluid Dynamics Laboratory (GFDL) hurricane model. The development of HWRF was a collaboration between the National Oceanic and Atmospheric
Administration / National Weather Service /The National Center for Environmental Prediction/ Environmental Modeling Center (NOAA/NWS/NCEP/EMC), NOAA GFDL, and the University of Rhode Island (URI). The HWRF model was launched in 2007 and has been used by the National Hurricane Center (NHC) for hurricane forecasts. Starting in 2011 the code was available to the general research community through the Developmental Testbed Center (DTC).



HWRF is composed of one outer domain 80° x 80° (18-*km* resolution) with grid spacing of 0.135° with two-way movable nested grids that follow tropical cyclones using an intermediate domain 12° x 12° (6-*km* resolution) with a grid spacing of 0.045° and cloud resolving domain 7.1° x 7.1° (2-*km* resolution) with a grid spacing of 0.015°. The physics parameterizations in the operational HWRF model are adapted to be specific to tropical cyclones. Some of the physics parameterization in the HWRF are:

GFDL surface physics (to account for the air-sea interaction over warm water and under high wind conditions), Rapid Radiative Transfer Model (RRTMG) with cloudiness parameterization, Ferrier-Aligo Microphysics, designed specifically for tropical cyclones, NCEP Global Forecast System (GFS) boundary layer, and Cumulus GFS Simplified Arakawa-Schubert (SAS) for deep and shallow convection. Other packages included in the model are: vortex initialization, prep-hybrid, which processes data in its original vertical coordinates, hybrid Ensemble Kalman Filter-Three Dimensional Variational (EnKF-3DVAR), Unified Post

Processor (UPP), and the GFDL vortex tracker.

The model initialization starts as soon as the NHC identifies a disturbance with potential for development into a tropical cyclone, which is called "invest," and has the potential to become a tropical cyclone in the near future. In order to obtain the center of the tropical cyclone the model is run. Output of this run is used as a first guess of the following run to identify the vortex. The lateral boundary conditions are obtained from the NCEP GFS forecast. The HWRF forecast is available every 6 hours in cycling mode

e.g. the previous 6 hour forecast is used to initialize the new HWRF forecast until the system becomes disorganized (e.g. due to: landfall, degeneration, or its transition to extra-tropical cyclone).

### 2.2 Lightning Parameterization

The LPI lightning parameterization estimates the potential for charge generation and separation that leads to lightning flashes in a convective thunderstorm between 0 °C and -20 °C (Yair et al., 2010). The most effective charge separation is driven by

collisions of ice and graupel particles in the presence of supercooled water (Yair et al., 2010; Lynn et al., 2012). The LPI is calculated by using simulated grid-scale vertical velocity and simulated hydrometeor mass mixing ratios of liquid water, cloud ice, snow, and graupel. The LPI is nonzero in the charging zone when the majority of the cells within a five-grid radius (10 *km*) of that grid point have a vertical velocity greater than 0.5 *m s$^{-1}$*. This corresponds to the growth phase of the thunderstorm (Lynn et al., 2012). The mathematical definition of the LPI is

$LPI = \frac{1}{V} \iiint \varepsilon w^2 \, dx \, dy \, dz$         (1)

where V is the volume of the air in the layer between 0 °C and -20 °C. The triple integral is computed within the cloud volume from the freezing level (altitude in *km* above the surface) to the height of the -20 °C isotherm; ε is dimensionless which has a value between 0 and 1 and is defined as

$\varepsilon = \frac{2(Q_i Q_1)^{0.5}}{Q_i Q_1}$         (2)

where $Q_i$ is the ice fractional mixing ratio in (*kg kg$^{-1}$*) defined as

$Q_i = q_g \left[ \left( \frac{(q_s q_g)^{0.5}}{(q_s + q_g)} \right) + \left( \frac{(q_i q_g)^{0.5}}{(q_i + q_g)} \right) \right]$         (3)

($q_s$) is the model computed mixing ratios (*kg kg$^{-1}$*) for snow, cloud ice ($q_i$), and graupel ($q_g$). $Q_1$ is the total liquid water mass mixing ratio in (*kg kg$^{-1}$*). In summary, ε is a scaling factor for the cloud updraft, and attains a maximum value when the mixing





ratios of supercooled liquid water ($Q_1$) and of the combined ice species ($Q_i$) are equal (Yair et al., 2010; Lynn et al., 2012). $w$ is the vertical wind component ($m\ s^{-1}$).

The LPI code was implemented into the Thompson microphysics scheme, which was already implemented in the HWRF model. The variables added to Thompson microphysics scheme in order to be able to calculate the LPI were: liquid mixing ratio, z_full
which is the area between t_base and t_top, where t_base=273.15 and t_top=253.15.

### 2.3 HWRF Idealized Framework

The Idealized tropical cyclone in the HWRF model in this study is set up as the 2015 operational HWRF (V3.7a ) with the exception that the sea surface temperature is maintained constant at 302 *K*. Initial conditions are generated from a vortex superposed on a base state inactive sounding. The initial wind speed maximum intensity of the vortex is set up as 20 *m s⁻¹* with a
radius of maximum winds of 90 *km* as described in Hurricane Weather Research and Forecasting (HWRF) Model: 2015 Scientific Documentation (Tallapragada et al., 2015).

Three experiments were conducted to evaluate and test the LPI implementation. The experiments were named Ideal_CTRL, Ideal_LPI and Ideal_LPI_newPBL (Phase 1 and 2). The Ideal_CTRL experiment used the Idealized HWRF with the unmodified Thompson microphysics scheme.   The Ideal_LPI experiment uses the same setup as the Ideal_CTRL but introduces the LPI
implementation into Thompson mucrophysics (Phase 1). The Ideal_CTRL and the Ideal_LPI simulations were performed lasting 126 hours. The third experiment named Ideal_LPI_newPBL was designed to examine eyewall replacement and lightning variability (Phase 2). This experiment is as Ideal_LPI but with a different planetary boundary layer scheme, deemed the "new PBL". This new PBL scheme is called The Hybrid Eddy Diffusivity Mass-Flux (EDMF) Boundary Layer Parameterization with Dissipative Heating and Modified Stable Boundary Layer (Han et al., 2015).  When using this new scheme in the Idealized
framework, and the simulation lasts more than six days, and an eyewall replacement cycle can be reproduced. When using the "new PBL", the eddy diffusivity mass-flux (EDMF) scheme (Siebesman and Teixeira 2000; Soares et al. 2004 and Siebesma et al., 2007) is used when the PBL has a strong instability and the eddy-diffusivity counter gradient mixing approach (EDCG) scheme (Deardorff 1966; Troen and Mahrt 1986; Hong and Pan 1996; Han and Pan 2011) is used for weaker instability of the PBL (Han et al., 2015). This new scheme helps to better forecast the convective boundary layer (CBL) in the tropics when the
instability is very strong (Han et al., 2015).

### 2.4 Model Verification and Validation for Real Cases

The World Wide Lightning Detection Network (WWLLN) data was used to verify the LPI forecast. The WWLLN data network contains over 70 sensors around the world. These sensors detect impulsive signals from lightning discharges called sferic in the very low frequency (VLF; 3-30 *kHz*). Even though these data was the best data we have at the moment for lightning around the
world this network has a very low detection efficiency of about 10.30 *%* in 2010 for cloud to ground lightning. For cloud-to-cloud lightning the detection efficiency is even lower (Abarca et al., 2011). As the number of sensors increases the detection efficiency will improve but still on the low side of detection efficiency. The intensity and track forecasts were verified with the best track hurricane data (HURDAT) which is a post storm reanalysis completed by the NHC for each tropical cyclone. These reanalyses are executed in terms of intensity, central pressure, size, and position and are available every 6 hours (Landsea and
Franklin 2012). Landsea and Franklin (2012) defined best track "as a subsequently smoothed representation of a tropical cyclone's history over its lifetime, based on post storm assessment of all available data."





### 2.5 Real Cases Synoptic Overviews

Two real cases have been evaluated using the HWRF with LPI implementation (Phase 3). These real cases are hurricanes Earl and Igor (2010). These two selected cases fit into the criteria of tropical cyclones that underwent rapid intensification.

### 2.5.1 Earl

Hurricane Earl developed from a strong tropical wave that exited from West Africa on August 23, 2010. A day later (August 24) the tropical wave developed a closed circulation. On August 25 convection activity increased, becoming tropical depression. Six hours later intensity increased to satisfy threshold requirements for becoming a tropical storm; Earl. As a tropical storm Earl moved westward to west northwestward in the eastern Atlantic under the influence of a strong subtropical ridge. During this time, tropical storm Earl was steadily intensifying due to favorable atmospherics conditions; e.g. weak to moderate shear and

warm sea surface temperatures. On August 29 Earl became a hurricane when it was located around 200 *n mi* from the northern Leeward Islands. On the same day Earl was able to start moving slower and more northward due to the weakening of the subtropical ridge caused by the passing hurricane Danielle. Earl began a rapid intensification cycle starting on August 29 until August, becoming a category 3 hurricane in the Saffir-Simpson Hurricane wind scale in less than 12 hours. The winds increased 40 *kt* in 24 hours becoming category 4 on August 30. An eyewall replacement cycle started around at the same time that Earl

reached Category 4 causing the period of rapid intensification to end, remaining with the same intensity over the next 24 hours.

During late August 31 and early September 1 Earl decreased is intensity to category 3 due to an increase in shear. Later that day the wind shear decreased, causing Earl to regain strength regaining strength to a category 4 hurricane and reaching its peak intensity of 125 *kt* when located 380 *n mi* southeast of Wilmington North Carolina. After reaching its maximum intensity, Earl started to rapidly weaken due to increased wind shear, colder sea surface temperatures, and drier air, becoming category 1

hurricane on September 3. Earl became a tropical storm on September 4 when was located 130 *n mi* south-southeast of Long Island, New York.

Earl restrengthened and became a hurricane category 1 on September 4 at 1200 UTC and make landfall 3 hours later (15500 UTC) in Liverpool, Nova Scotia Canada. A second landfall as a tropical storm was recorded at Prince Edward Island at 1900 UTC (Cangialosi 2011).

### 2.5.2 Igor

Igor developed from a strong low-pressure system that exited West Africa on September 6, 2010. Two days later (September 8) organized convection close to the low pressure developed and formed a tropical depression about 80 *n mi* southeastern of the Cape Verde Islands. Within 6 hours, tropical storm Igor formed but weakened due to an interaction with a disturbance shortly after. On September 10, atmospheric conditions became more favorable and Igor strengthened from a tropical depression to a

tropical storm as it was moving westward. A steady intensification occurred and Igor became a Hurricane on September 12 followed by a rapid intensification on September 13. On September 14 Igor decreased intensity but later intensified reaching its peak intensity on September 15 as it started to move west northwestward direction. Once Igor reached its peak intensity, an eyewall replacement cycle began causing the hurricane to decrease in strength. The eyewall replacement cycle was over by September 16 and Igor regained strength. After September 17 Igor continued north encountering unfavorable atmospheric

conditions for tropical cylones. During this steady weakening Igor increased its size considerably and tropical storm winds force extended to 300 *n mi* from the center. As Igor was moving north northwestward, it accelerated and increased in size (approx. 750



*n mi*). Igor made landfall in Cape Race Newfoundland. Upon exiting Newfoundland, the extratropical transition began on September 21.

## 3 Results

### 3.1 Idealized Case Studies

Time series analyses of maximum wind speeds for the Ideal_CTRL and the Ideal_LPI experiments were performed to verify that the implementation was performed correctly (phase 1). The idealized tropical cyclone was run with an HWRF version, with and without the LPI implementation. As expected, the results from the Ideal_CTRL experiment showed that the implementation did not affect the model forecast, as the LPI is a diagnostic tool.

An LPI time series was calculated by averaging LPI over each grid point represented in Figure 1 as the gray solid line. Results

from this time series were used to evaluate the relationship between the forecasted maximum wind and the LPI. The LPI time series exhibits an oscillatory behavior. The first maxima in LPI occurred during the period of intensification (forecast hour 20 with an LPI maximum of 6,596 $J\ kg^{-1}$). The second maxima and highest value of LPI 9,699 $J\ kg^{-1}$ occurred at forecast hour 51, and the third maxima occurred at forecast hour 77 with an LPI value of 6,931 $J\ kg^{-1}$. The mean behavior over this simulation exhibited a positive correlation of r = 0.65 between wind speed and LPI during the period of intensification (Figure 1). This

behavior changes during the peak intensity period (forecast hours 100-120). Consistent with observations, our results show an increase in LPI prior to wind intensity maximum (Price et al., 2009).

These results are in agreement with previous findings by Price et al., 2009 (e.g. lightning peak occurs hours before the intensity peak). Further evaluation of the LPI simulation in Figure 2 shows that when the tropical cyclone is weaker (forecast hour 52), there is greater potential for lightning to occur in the outer rain bands and simultaneously the inner core (Figure 2). When the

tropical cyclone is mature (forecast hour 110), the potential for LPI is concentrated in the inner core. These results coincide with observations reported by Molinari et al., (1999). At forecast hour 52 (Figure 1) one can note that an LPI maximum of 9,329 $J\ kg^{-1}$ corresponds to maximum winds speed of 48 $m\ s^{-1}$. At this time in the simulation the tropical cyclone was undergoing steady state intensification and the potential for lightning was located in its majority in the outer rain bands (Figure 2). At the simulation time of hour 110 (Figure 2) the tropical cyclone had reached its maximum winds speed of 68 $m\ s^{-1}$ (Figure 1). At this time, the

LPI values are much lower (2,232 $J\ kg^{-1}$; Figure 1) and the potential for lightning is located near the inner core (Figure 2).

LPI results were compared against model-predicted radar reflectivity from model output since radar reflectivity values can provide information about cloud electrification that leads to lightning. Results from this analysis showed that the reflectivity resembles the same behavior as the LPI. High reflectivity values (e.g., greater than 40 *dBZ*) were observed in the inner core and in the outer rain bands during the period of intensification (Figure 3). When the idealized tropical cyclone reached intensity peak

high reflectivity, values were observed in the inner core with no reflectivity values in the outer rain bands. When comparing locations of high reflectivity with the LPI results shown, the locations of higher LPI are the same as the high values of reflectivity (Figures 2 and 3).

### 3.2 Idealized Case Study with Eyewall Replacement Cycle

The characterization of lightning activity before, during, and after an eyewall replacement cycle has been particularly

challenging due to lack of direct observation of eyewall replacement. Since an eyewall replacement cycle is observed in this Idealized case, an analysis of this relationship was performed (phase 2).



A time series of LPI values against maximum winds speed was plotted in Figure 4. The LPI forecast shows a steady increase in LPI and maximum winds but without consistent positive or negative correlations. The relationship between lightning and intensity changes observed in previous cases e.g. an increase in lightning activity correlates to a weaker or less intense tropical cyclone and less amount of lightning corresponds to a stronger tropical cyclone or an intensity peak was not that evident in this

experiment (Figure 4). In this experiment, it was also found no conclusive evidence for correlation given the weak positive correlation of r = 0.29 during the eyewall replacement cycle.

To be able to observe and corroborate the eyewall replacement-cycle in this experiment, a cylindrical coordinated transformation was executed. After the model outputs results were transformed into cylindrical coordinates the variable mean azimuthal tangential winds were calculated and plotted (Figure 5).

Contour plots of the mean tangential winds as a function of altitude and distance from the center of circulation are presented in Figure 5. Each one of the eight plots (a-h) represents a 6 hour time step during the forty-two hour window in which the eyewall replacement occurred. This eyewall replacement started at forecast hour 130 and ended at forecast hour 172.  In Figure 5a the tangential wind maximum is located at a radial distance with maximum winds between 50 and 70 *km*. In Figure 5b, 5c, and 5d the area of maximum tangential winds has elongated, covering more distance at lower heights (e.g. from 50 *km* to more than 100

*km*). In Figure 5e and 5f two different locations of tangential winds maxima are observed. In these two Figures there is a second region with a tangential winds maximum located approximately at 100 *km* radius and that the first are of tangential wind maximum, and is weaker than the second area of tangential winds maximum. Six hours later (Figure 5g) the Idealized tropical cyclone has a tangential wind maximum located at a radius beyond 100 *km*. At this time in the forecast the second area of tangential wind maximum replaces the original area of tangential winds maximum.

The second area of tangential winds maximum cuts the heat supply into the primary eyewall causing its decay (Bell et al., 2011). This new eyewall (area of tangential wind maximum) is located in a larger radius of maximum winds (RMW) i.e. beyond 100 *km*. In Figure 5h the eyewall begins to contract and move to a smaller RMW over 80 *km*. This new eyewall started to contract but remains larger than the original eyewall.

The forecasted composite radar reflectivity and the average LPI over each grid box were analyzed for the same period, shown

for the eyewall replacements cycle (e.g., forecast hour 136- 172) every six hours (Figure 6). In Figure 6, the shaded color represents LPI values and the contoured black lines represent composite radar reflectivity from -5 to 60 *dBZ* every 5 *dBZ* for the 48 hours of the eyewall replacement. The development of a secondary eyewall can be seen over the eyewall replacement period, even though it does not show perfectly concentric double walls or rings. A significant amount of LPI was forecast in the outer rain bands during the entire eyewall replacement cycle but with a slight decrease toward the end of this period. This is consistent

with the behavior observed in the Ideal_LPI experiment.

At the cross section of 24° N latitude, the vertical velocities were analyzed for the same time period, as shown in Figure 6 (e.g., forecast time showing eyewall replacement cycle).  In figure 7 the positive numbers e.g. $0 - 5\ m\ s^{-1}$ are indicative of updrafts and the negative numbers 0 – (-2) are indicative of downdrafts. In this figure, it is also shows where 0, -10, and -20 °C isotherm is relative to height, as well as the reflectivity (countered), for 5 purple (contour) and 30 *dBZ* (blue contour). In these cross sections,

it shows that at forecast hour 130, there are no significant downdrafts. Six hours later, some small areas of downdrafts are observed in the outer region of the tropical cyclone. As forecast time increases, the areas of downdrafts increased, reaching the strongest downdrafts at forecast hour 154. The locations of the strong downdrafts are distributed from the inner core through the





outer region. After forecast hour 154, the downdrafts decreased and were mainly located in the outer bands. Additionally, strong updrafts are observed and can be associated with the double inner core structure. Weaker updrafts were located near the inner core (e.g., 1-3 $m\ s^{-1}$) and stronger updrafts (e.g., 4 -5 $m\ s^{-1}$) are further from the center of the tropical cyclone. This can be observed particularly in the forecast hours of 154 and 160.

### 3.3 Earl

A 126-hour simulation with outputs every hour was performed e.g. August 28, 2010 0000 UTC – September 2, 2010 0600UTC. The simulation forecast hours were selected to capture: steady and rapid intensification, including the maximum winds of Earl.

In Figure 8 a time series between HWRF forecasted LPI and maximum winds as well as the maximum winds from best track is shown. Where the solid black line is for LPI ($J\ kg^{-1}$), the red line is for the HWRF maximum winds ($m\ s^{-1}$) and the dashed blue line is best track maximum winds ($m\ s^{-1}$). An increasing trend in potential for lightning is observed during the first 60 hours of the simulation as Earl was intensifying with a positive correlation of r = 77. A negative correlation of r = -0.79 was forecasted with a maximum LPI peak of 17,535.55 $J\ kg^{-1}$ occurring 22 hours prior to the intensity peak forecasted by HWRF (forecast hour 30). At the time when HWRF forecasted an intensity peak of 59 $m\ s^{-1}$ the potential for lightning had decreased to a local minimum of 11,115.42 $J\ kg^{-1}$. At forecast hour 105 the potential for lightning increased to 17,734.42 $J\ kg^{-1}$, but this increase in LPI did not correlate to any increase in HWRF maximum winds according to the HWRF forecast. Results from this simulation show that the LPI negatively correlates to maximum winds (Figure 8). The observed relationship is that LPI peaks before HWRF maximum wins peak and further, that at the maximum wind peak a local LPI minimum is observed e.g. negative correlation.

A comparison between the HWRF forecasted storm intensity against best track intensity from the NHC it is also shown on Figure 8. In Figure 8, it is observed that the model captured and forecasted the observed rapid intensification. Furthermore, the model failed to forecast the second period of intensification that started at 101 hours into the forecast time and persisted until the end of the simulation.

When comparing the LPI values with the best track maximum winds values from NHC (Figure 8), one can see that the LPI increased as Earl was intensifying (forecast hour 0-60). During forecast hours 65 – 90 Earl maximum winds remain relatively constant and during this period a decrease in LPI was observed e.g. the end of rapid intensification. Between the forecast hours 101 and 105 and increase in LPI values with the last peak of 18,028.71 $J\ kg^{-1}$ was observed. Following this LPI peak e.g. 101 – 105 Earl's peak intensity was observed.

The WWLLN data (black dots) and the LPI values (shaded colors) were plotted alongside for all simulation times but the results shown in Figure 9 only covers 48 hour period, every six hours between August 29 0000 UTC and August 31 0000 UTC. This period was chosen for analysis due to the small track error between HWRF (red line) and best track (black line; Figure 10). Even though the full analysis of the track forecast is beyond the scope of this study it needs to be analyzed for validation of the LPI location versus lightning observation (WWLLN data). The locations of the lightning observed are offset with the LPI values due to a half-degree discrepancy between the observed track and the track forecasted by HWRF (Figure 10). From forecast hours 0000 UTC –0200 UTC on August 29 most of the lightning is located in the eastern and southern quadrants. After forecast hour 0300 UTC most of the lightning activity was observed in the left quadrant of Earl mostly in the outer rain bands. Lightning activity decreased during the period 0800 – 1100 UTC. Then over the next several hours lightning was redistributed to the rain bands. In August 29 1800 UTC two strong rain bands with high amount of lightning are observed. This strong rain band with a large amount of lightning persisted for about 11 hours. After the lightning activity decreased in the rainband lightning activity





increased tremendously at August 30 at 0400 UTC in the southern quadrant of Earl. Four hours later (August 30 at 0800 UTC) lightning was observed on the rain bands. After August 30 at 1800 UTC the lightning activity was scattered and was primarily located in the outer rain bands. In general, if the track forecasted by HWRF is shifted a half-degree south and in sometimes even less than that, locations of lightning and maximum values of LPI results are in agreement. In Figure 9 it is also seen that when

the tropical cyclone was organizing and intensifying, the lightning location was observed in the outer rain bands. These results coincide with the hypothesis by Price et al., 2009 and Molinari et al., 1998.

The rate of lightning per hour for hurricane Earl was calculated using the WWLLN data and compared to the LPI averaged per hour for the entire domain (Figure 11). On Figure 11 the black line represents the LPI and the blue dashed line represents the observed lightning per hour from the WWLLN data. Results from Figure 11 shown that the LPI has potential to forecast the

observed lightning peaks. Based on the observations from WWLLN there was an increase in lightning activity (1,538 lightning per hour) forecast hour 30. A second and more intense lightning outbreak (1,617 lightning per hour) occurred at forecast hour 54. The third lightning peak of 1,102 lightning per hour occurred at forecast hour 103. For each of these particular observed lightning increases an LPI increase was forecasted as well.

### 3.4 Igor

The simulation for Igor was initiated on September 10 at 0600 UTC with the purpose of capturing the rapid intensification period and the peak intensity. A time series of hurricane Igor was performed as shown in Figure 12 The solid black line is for the LPI ($J\ kg^{-1}$), the red line is the maximum winds ($m\ s^{-1}$) forecasted by HWRF and the blue shaded line is the maximum winds ($m\ s^{-1}$) from the best track. Results from this analysis show a positive correlation of r = 0.82 during the period of intensification e.g. forecast hours 0 -70 and a negative correlation of r = -0.29 between HWRF wind speed and the maxima of the LPI during the

peak intensity. HWRF forecasted the first increase in LPI and maximum winds at hour 20. Steady intensification was forecasted as well as an increase on LPI after hour 40. The strongest LPI peak of 20,610 $J\ kg^{-1}$ was forecasted at hour 85, which was during a minimum in intensity and 10 hours prior to the predicted intensity peak. A negative correlation trend of LPI starts after Igor forecasted intensity peak of 48 $m\ s^{-1}$. Fifteen hours after the intensity peak was forecasted another increase in LPI e.g. 16,114 $J\ kg^{-1}$ was observed.

When comparing the LPI forecast against the best track (Figure 12) e.g. the solid black line is the LPI forecast and blue dashed line is best track from NHC, during the period of rapid intensification e.g. forecast hour 16 – 65 a positive correlation of r = 0.92 between LPI and best track maximum winds is observed. This time series shows that the most intense LPI peak of 20,610 J kg $^{-1}$ occurred 30 hours prior to the observed intensity peak of 69 $m\ s^{-1}$. During the intensity peak Igor's wind speed maxima, a LPI maxima was also observed, showing a positive correlation of r = 0.59.


The forecasted LPI and the WWLLN where overly plotter for validation (figure not shown). According to observations from the WWLLN data the majority of the lightning occurred after the period of rapid intensification. The higher lightning activity was observed after the forecast hour 100 which is 15 hours prior to Igor peak intensity. Most of the lightning activity occurred in the southeastern quadrant of hurricane Igor and in the outer rainbands with no lightning activity in the inner core. It is important to

mention that there were few reports of lightning in WWLLN data for this particular case which make this validation hard to accomplish. On the other hand, the LPI forecasted location coincides with the low amount of observations available.





The lightning per hour from the WWLLN data was also compared to the LPI averaged over the entire domain forecasted by HWRF (Figure 14). Results from this comparison shown that the time periods where more lightning was observed are not exactly the same time for the LPI peaks. At the first hour of the forecast a LPI peak was observed but observations shown that a lightning peak occurred four hours later. During the forecast time 25-65 there is not much lightning observed with many hours

5 with non lightning observations. At forecast hour 80 an LPI peak coincides with an observed lightning peak. After forecast hour 100 the pattern is the similar between the forecasted LPI and the observed lightning.

## 4 Discussion

This investigation described the implementation of the LPI lightning parameterization scheme into the operational HWRF model and results from an examination of the relationship between lightning fluctuations and intensity changes in an Idealized

10 framework as well as real cases. The ultimate goal of this investigation was to find whether using lightning as proxy for tropical cyclone intensity changes had operational research value. Our hypothesis was that lightning could be a diagnostic tool to verify tropical cyclone intensity forecasts.

This study was executed in three phases. The first phase involved the implementation of the LPI code into HWRF and testing this implementation using an Idealized tropical cyclone simulation. The second phase of this study was designed to test the LPI

15 implementation with an Idealized frame work that reproduced an eyewall replacement and the third phase was LPI testing and evaluation with real cases. Two real case studies were tested; Earl and Igor. These two real cases fit the criteria of TCs that underwent rapid intensification and were category 3 or more on the Saffir-Simpson tropical cyclone scale.

This is the first time that the lightning and intensity forecast correlation has been evaluated in an operational model. Our results support observations that a lightning maximum occurs prior to the intensity peak. Additionally, our results predict generally a

20 decrease in lightning potential during wind speed maxima. Our results also indicate that developing tropical cyclones should have a broader spatial distribution of the potential for lightning and that the majority of the lightning is located in the outer rain bands. In contrast, more mature tropical cyclones exhibit a lower a potential for lightning and it is located in the inner core. It is also important to mention that the LPI location in all experiments coincides with high radar reflectivity values as well an eyewall replacement was successfully simulated. However, for the eyewall replacement experiment, the relationship between LPI and

25 intensity changes is inconclusive.

The increase in LPI prior to the increase in intensity was captured in the Ideal_LPI experiment. In these results, the timing of the LPI increase and the ensuing increase in intensity were not observed as in previous results. In previous results, a lightning peak was observed 30 hours prior to intensity peak. In this investigation, the second and larger LPI maxima occurred 60 hours prior to maximum intensity (Figure 1). In the Idealized simulation with eyewall replacements the LPI showed a lagged correlation in

30 which both the LPI and intensity continuously increased over time (Figure 4).

The Idealized tropical cyclone experiment executed with the new PBL scheme (EDMF) shows that by using this new PBL scheme, the model has the capability of producing eyewall replacement. In this experiment, the LPI correlates to maximum winds but with a tendency to increase over time (Figure 4). To better understand why the LPI keeps increasing over time, vertical velocities cross-sections were analyzed (Figure 7). Results from this analysis showed that the updraft was strong during

35 the eyewall replacement cycle, explaining why the LPI values also kept increasing over time (as shown in Figure 7). The LPI calculations are directly proportional to vertical velocities as they are calculated in another subroutine within the model (equations 1 and 2).





Results from each real case study were compared with the WWLLN data and the best track data from NHC. The WWLLN data is limited by the large amount of lightning that is not reported in the network due to its low detection frequency especially on cloud-to-cloud lightning (Abarca et al., 2011). This could explain the low amount of lightning observed, in particular, for hurricane Igor. The WWLLN data was the best data available at present of the year 2010.

Of the two hurricanes selected for study, the relationship, positive and negative correlations between lightning and intensity changes were most evident for hurricane Earl. This was likely due to the large amount of lightning recorded by WWLLN (more than 48,000 lightning; Stevenson et al., 2014). It is also important to mention that if we assume the relationship between lightning and maximum winds allows for lightning to be used as a proxy for intensity changes, then LPI could have been used as a warning for a possible error in HWRF intensity forecast. An LPI peak was forecasted after the intensity peak in Earl (Figure

8). Subsequent observations from NHC showed a second intensity peak that HWRF failed to forecast (Figure 8).

The relationship between lightning and maximum winds was not clear in the last 26 hours of the simulation for Igor (Figure 12). Results from Igor could be associated with the second theory proposed by Thomas et al., 2010; DeMaria et al., 2012 where an increase in lightning activity in the inner core was associated with the end of intensification (Figure 12).

Radar reflectivity is another diagnostic tool that is useful for evaluating the intensity of the convection and has also been used for

assessing lightning distributions. Results from the evaluation of the reflectivities calculated by HWRF for each case study show that not all of the areas with high reflectivity values were necessarily associated with lightning observations. When the LPI was compared with the radar reflectivity across these two cases, a one-to-one relationship was not observed. In many times, areas with high reflectivity did not necessarily contain a large potential for lightning. The expectation that LPI would correlate well with radar reflectivity (due to the latter's association with lightning) was not confirmed. Generally speaking, the higher the

observed reflectivity value the more intense is associated updraft and the ensuing charge separation leads to lightning formation. The same is true with the lightning data from WWLLN. The lightning locations do not necessarily match up with the areas of high reflectivity. They do, however, tend to be associated with the potential for lightning. High values of LPI do not necessarily mean that a lightning strike will occur, it means that the potential for a lightning to happen has increased. The forecasted LPI appears to be a more reliable tool for forecasting the possible locations of lightning with the correct storm location.

Previous studies present cases for the lightning peak occurring both before and after intensification. The results of this study confirmed that both scenarios are indeed common. The results from these simulations are statistically similar to the results obtained by Price et al., 2009 in their analysis of lightning observations for 56 TCs in all ocean basins. A lightning peak was observed on average 30 hours before the intensity peak. Even though only two hurricanes were analyzed, the negative correlation results are consistent.

In the case of hurricane Earl, the LPI was a predictor of an increase in intensity and in the hurricane Igor case, the LPI peak was a predictor of the end of intensification. These results are consistent with the outcomes of studies based on empirical observations. This study underscores the need to account for the prediction of lightning while forecasting the intensity on a tropical cyclone. In every instance, there is a fluctuation in the potential for lightning e.g. an increase or decrease on LPI and the intensity in TCs either increases or decreases.

The inability of the code implemented in this study to predict flash density, is be a critical shortfall. Flash density is defined as the cloud to ground flash rate integrated over time, which accounts for the flashes per area (Makela et al., 2011). Currently WRF-ARW has other lightning parameterizations that can forecast explicit charges and discharges, but the implementation of explicit



microphysics is required for this scheme (e.g. the NSSL two- moment); which is not available in HWRF operational at the moment. These shortfalls demonstrate there is still much work to do to be able to forecast lightning flash density in numerical models in particular HWRF.

On November 2016 NOAA and NASA successfully launched GOES-16. This new satellite has one of the first Geostationary Lightning Mappers (GLM) on board. The GLM will provide the capability to measure total lightning including cloud-to-cloud and cloud-to-ground. Data will be available globally with a spatial resolution close to 10 *km*. This new real time lightning data will help to elucidate the relationship between lightning and TCs with the ultimate goal of improving intensity forecast.

**5 Conclusions**

This investigation is the first to forecast lightning using an operational version of HWRF. The existence of a relationship
between lightning and changes in intensity in TCs was documented using an Idealized framework and two real case studies. The results for the correlations between lightning and storm intensity changes were found to be consistent with previous studies, based on observations.

Regarding the leading questions of this study, the answers found here are as follows: (i) Can the HWRF model predict lightning spatial-temporal distributions with an acceptable degree of accuracy? Lightning location observations (WWLLN data) were
compared with the LPI for the rapid intensifying TCs. Despite small track errors (Figures 10 and 13) for hurricane Earl, and the low detection efficiency of the WWLLN data for hurricane Igor, the HWRF with the LPI implementation forecasted the lightning location with acceptable accuracy.

(ii) How well does the HWRF with lightning parameterization model forecast lightning spatial distributions before, during, and after tropical cyclone intensification? A positive correlation between LPI and maximum winds was found during the period of
intensification for hurricane Earl and Igor (Figure 8 and 12). Followed by a negative correlation during the period of peak intensity with the exception of a positive correlation between LPI and maximum winds for hurricane Igor in the last hours of the simulation.

(iii) What is the functional relationship between tropical cyclone wind speed and lightning frequency in the HWRF model forecast? The HWRF 2015 (V3.7a) operational version consistently predicted a peak on the potential for lightning prior to
maximum winds peak with an average of 22 hours before intensity peak for real cases. The functional relationship between tropical cyclone wind speed and lightning frequency was the same reported in previous studies based on observation e.g. a negative correlation during the peak intensity of the tropical cyclone.

The results obtained from testing the LPI parameterization scheme in HWRF indicated that the lightning potential increase was always related to a change in intensity. Lightning fluctuations were found to be correlated with intensification and intensity
decay.

The availability of new lightning data available from the GOES-16 satellite will provide new opportunities for data assimilation and modeling. GOES-16 lightning data promises to have a better detection efficiency which will enable future work in this area. Enhanced detection efficiency is essential for a better understanding of lightning outbreaks in tropical cyclones. Our findings in this investigation demonstrate the potential for an important role of lightning as a diagnostic tool during the life cycle of tropical
cyclones.





Code Availability

The HWRF System components - V3.7a (August, 2015) is available at https://dtcenter.org/HurrWRF/users/downloads/index.php
To access the LPI implementation into the Thompson microphysics and the HWRF code please use the files located at
https://zenodo.org/badge/latestdoi/186898012

Author contribution

Keren Rosado: Implemented the model code to be tested. Designed and performed all the experiments as well as prepared the
10    manuscript with feedback from all of the co-authors.

Vernon Morris: Conceptualization and editing of the manuscripts. Funding acquisition

Vijay Tallapragada: Provided computational resources to accomplish the investigation as well and feedback on the progress.

Lin Zhu: Guided the first author to accomplish the code implementation.

Bin Liu: Helped post processing of observations to validate experiments results.

Competing interests

The authors declare that they have no conflict of interest.

20    *Acknowledgments* This material is based upon Keren Rosado supported by the National Oceanic and Atmospheric
Administration, Educational Partnership Program, U.S. Department of Commerce, under Agreement No. **NA11SEC4810003.**
Any opinions, findings, conclusions, or recommendations expressed in this publication are those of the author(s) and do not
necessarily reflect the view of the U.S. Department of Commerce, National Oceanic and Atmospheric Administration.



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





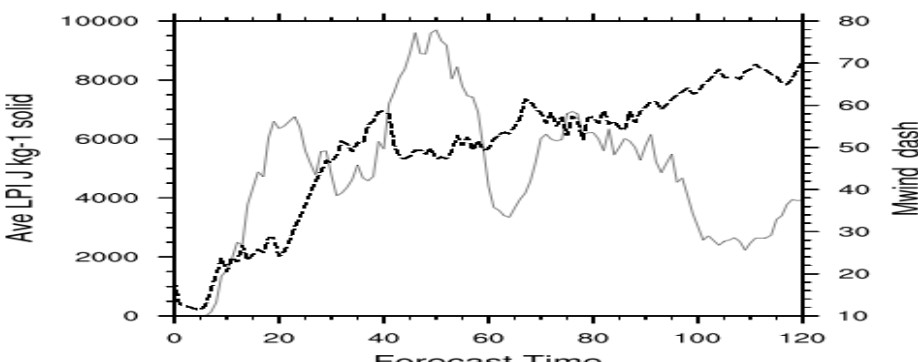

Fig.1. LPI values (gray solid line; *J kg⁻¹*) versus the wind speed maxima (black dashed line; *m s⁻¹*) calculated in the Ideal_LPI

10 experiment.

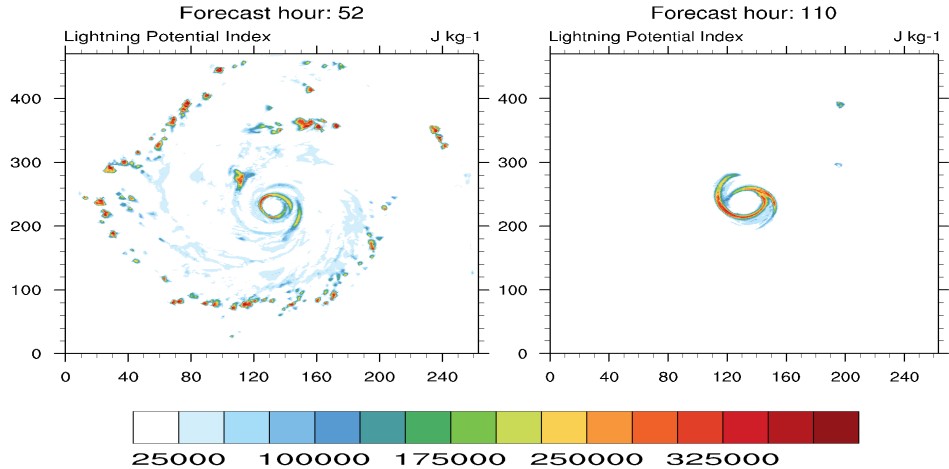

Fig.2. Lightning potential index (LPI) shaded, grid dx and dy: maximum LPI values (forecast hour 52) and minimum LPI values

20 (forecast hour 110) forecasted in the Idealized framework.





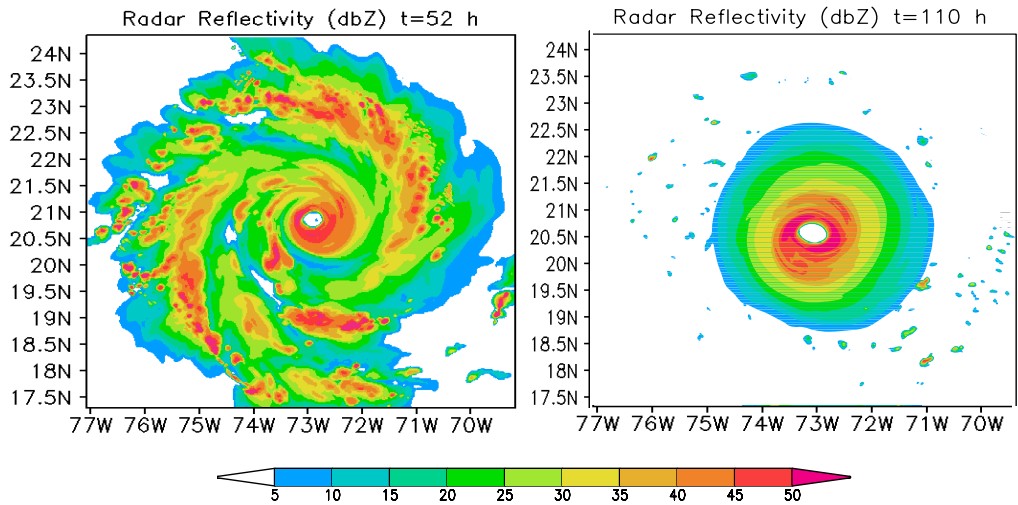

10    Fig.3. Radar reflectivity (*dbZ*) for the Idealized tropical cyclone with the LPI implementation forecast hour 52 and 110.

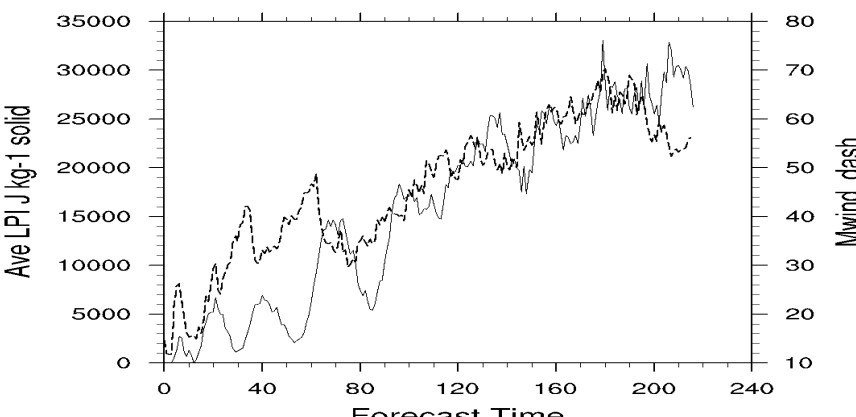

Fig.4. The solid line is the LPI (*J kg$^{-1}$*) values and dashed line is the maximum wind speed (*m s$^{-1}$*) for the nine days simulation of

20    the Idealized case with eyewall replacement cycle.





Fig.5. Height in *km* versus radius of maximum winds (*km*) plots contoured mean tangential winds (*m s⁻¹*) during the eyewall replacement for the Idealized case with the new PBL scheme.   Only showing forty-two hours of simulation every six hours. Note: the eyewall replacement cycle total time was nearby 48 hours.





CONTOUR FROM -5 TO 60 BY 5

25   Fig.6. LPI (shaded) *J kg⁻¹* and Composite Radar Reflectivity (contours) *dbZ* forecasted by HWRF during the eyewall
replacement e.g. from forecast hour 130 – 172 every six hours.







Fig.7. Cross section at latitude 24 °N during the eyewall replacement. The purple line represents the location of 5 (*dbZ*); blue line represents the location of 30 (*dbZ*). The shaded colors represent the vertical velocities (*m s$^{-1}$*) and the dashed grey lines contour temperature for 0° C, -10° C, and -20 ° C.

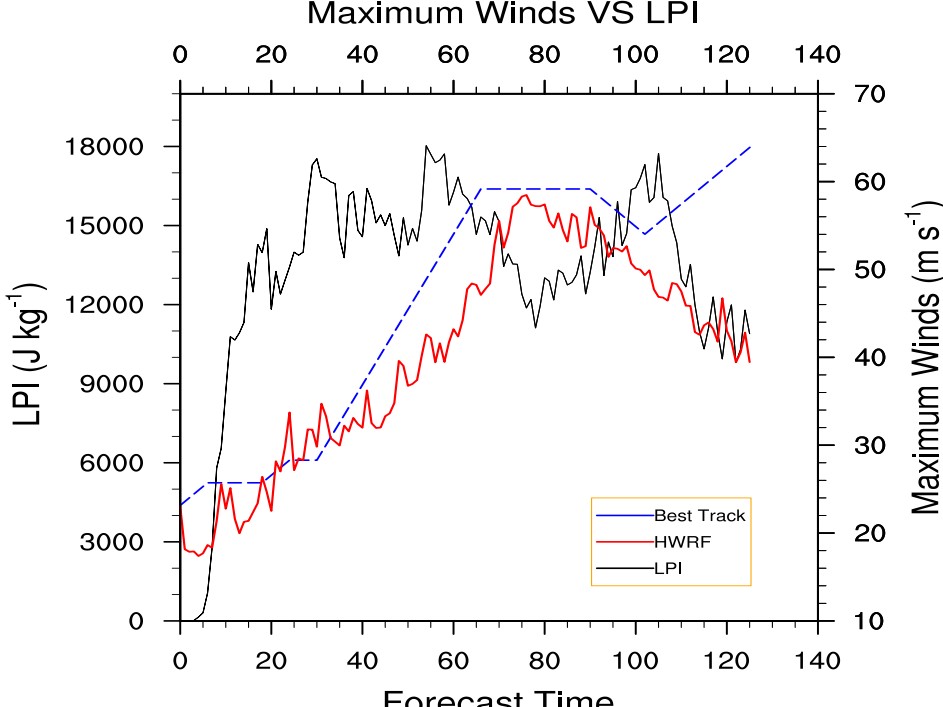

Fig.8. A 126-hour maximum winds and LPI time series from 2015 HWRF (V3.7a ) forecast and the maximum winds best track from the NHC for hurricane Earl 2010. Solid black line is the LPI in *J kg$^{-1}$*, the red line is the HWRF maximum winds speeds in *m s$^{-1}$* and the blue shaded line is the best track maximum winds in *m s$^{-1}$*.



Fig.9. Every six hours HWRF forecast of LPI in $J\,kg^{-1}$ (color shaded) for hurricane Earl for the period between August 29 0000 UTC and August 30 1800 UTC. Overlying is the corresponding total lightning observations from the WWLLN (black dots).



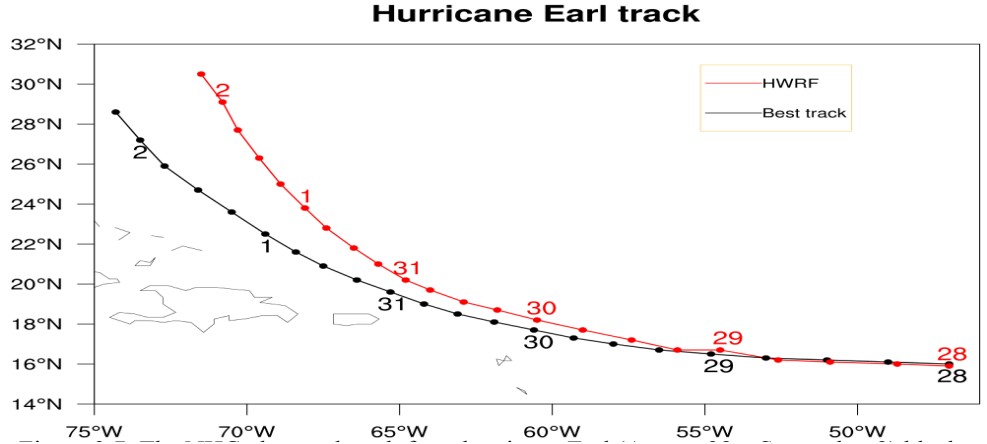

Fig.10. The NHC observed track from hurricane Earl (August 28 – September 2) black line with dots every 6 hours. The
forecasted track by HWRF 2015 (V3.7a ) for August 28 – September 2 red line with dots every 6 hours.

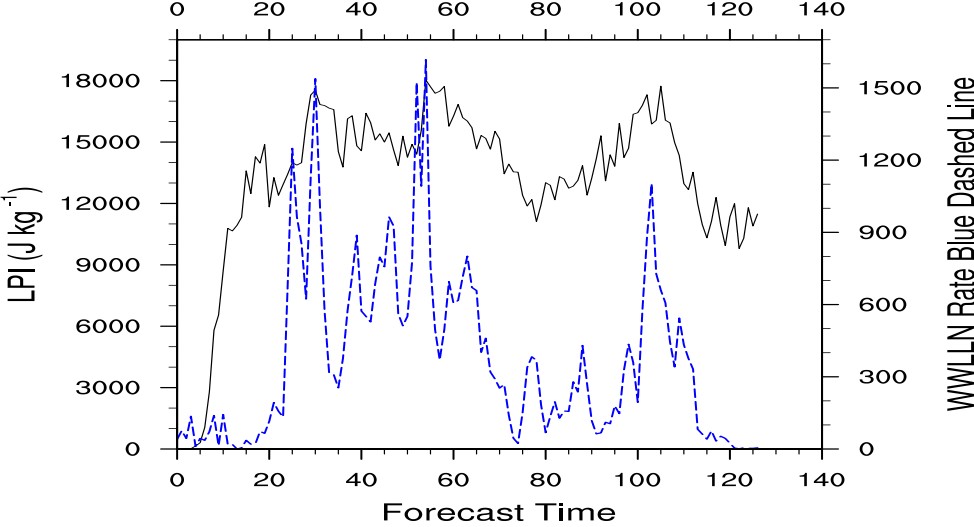

Fig.11. The rate from the WWLLN hourly lightning for hurricane Earl (dashed blue line) and the LPI ($J\ kg^{-1}$) averaged over the
domain per hour (solid line).



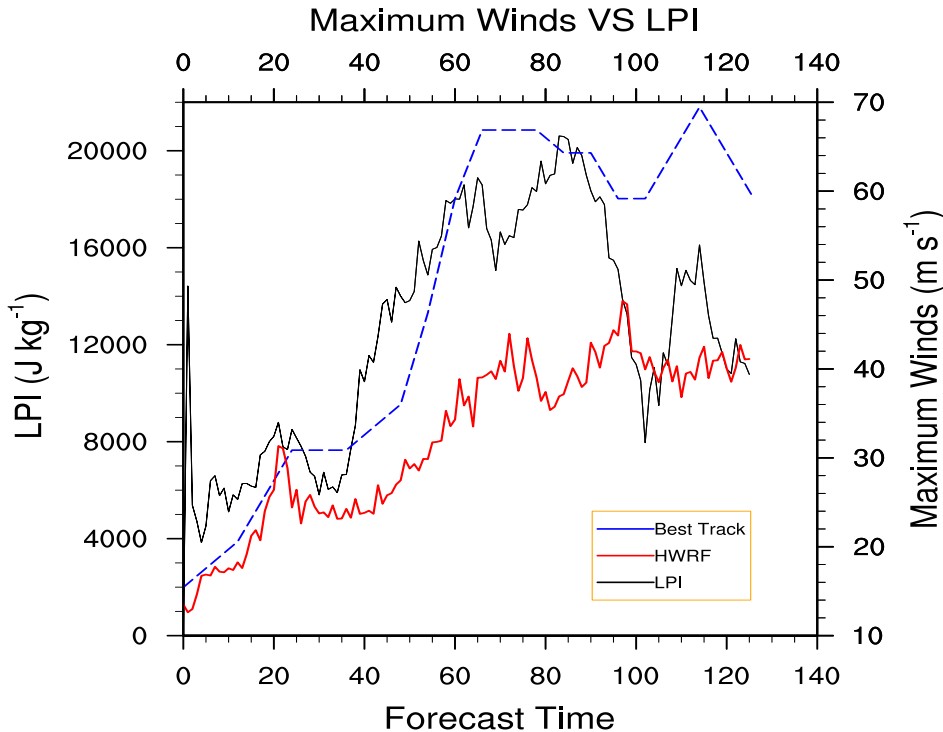

Fig.12. A 126-hour maximum winds and LPI time series from 2015 HWRF (V3.7a ) forecast and the maximum winds best track from the NHC for hurricane Igor 2010. Solid black line is the LPI in *J kg $^{-1}$*, the red line is the HWRF maximum winds speeds in *m s $^{-1}$* and the blue shaded line is the best track maximum winds in *m s $^{-1}$*.

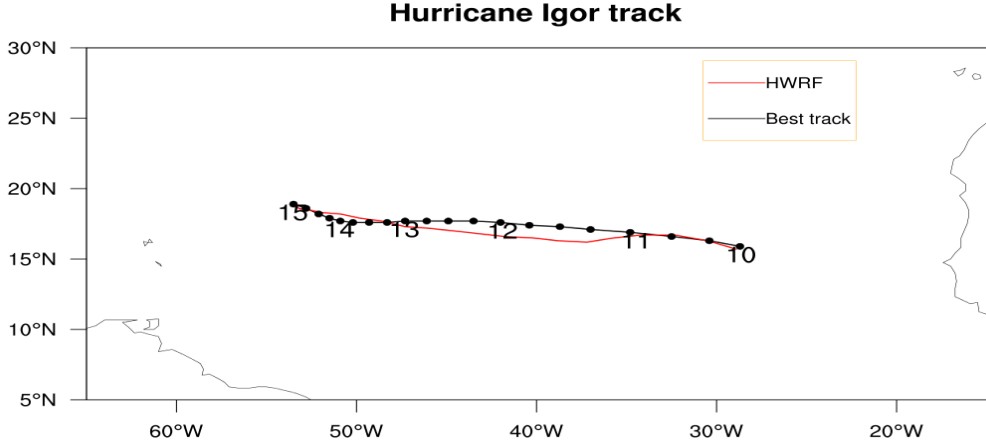

Fig.13. The NHC observed track from hurricane Igor (September 10 – September 15) black line with dots every 6 hours. The forecasted track by HWRF 2015 (V3.7a ) for September 10 – September 15 red line with dots every 6 hours.



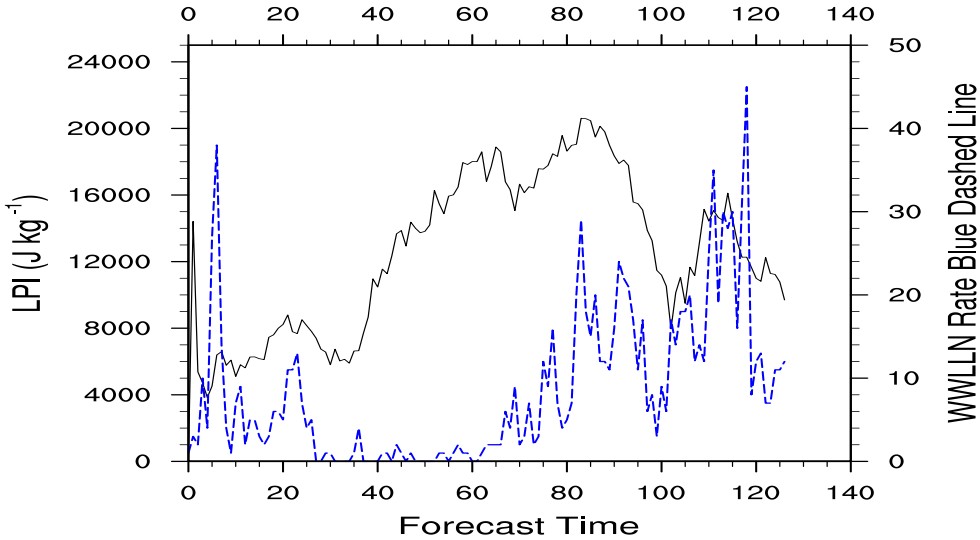

Fig.14. The rate from the WWLLN hourly lightning for hurricane Igor (dashed blue line) and the LPI ($J\,kg^{-1}$) averaged over the domain per hour (solid line).