# Peer review of "Spatial and Temporal Evolution of a Lightning Diagnostic in HWRF (V3.7a)"

_Geoscientific Model Development, 2019_

## Referee Comment (RC1) · Anonymous Referee #1 · 16 Aug 2019

Review of gmd-2019-139: Title: "Spatial and Temporal Evolution of a Lightning Diagnostic in HWRF (V3.7a)." by Rosado et al.

Summary: The authors utilize a diagnostic lightning forecast module to diagnose/estimate a measure of lightning activity within selected real tropical cyclone cases and an idealized case scenario.

Recommendation: Reject and, eventually, re-submit.

Main/General Comments: I. While the topic at hand is of interest to the community, I found the analysis generally very rudimentary with the authors going at length in describing figures limited to simple time series and horizontal/vertical cross sections. Because several tropical cyclone (TC) cases were simulated, the same basic anal-

ysis is repeated in a redundant manner, which "adds to the injury". After reading a few pages, I honestly got bored. Given its repetitive nature, the entire results section could easily be condensed into 2-3 pages with the content distilled into concise arguments/hypotheses. With this subsequent gain in text length, the analysis could then be expanded by including more elaborated means to analyze in more depth the lightning and microphysics in the present hurricane simulations. Examples of such analyses are actually provided in the (very few) existing explicit modeling studies of (2D or three dimensional branched) lightning within TCs such as Fierro and Mansell (2017, 2018) none of which are actually either referenced nor discussed to put this study into a more appropriate context. In addition to the usage of a very basic, 100% diagnostic lightning scheme (i.e., no explicit storm electrification physics), the idealized TC (or "donut storm") in Figs. 2 and 3 at 110h is completely unrealistic and, as such, cannot be used for verification. There are no rainbands and the eyewall width and eye diameters are unrealistically large. In the light of these unrealistic results alone, I am unfortunately inclined to recommend an editorial decision of rejection, for the time being. Additional major issues are listed below:

II. One salient concern of this study is the lack of rigor in analyzing TC lightning with respect to the inner core ($\sim$ r= 0-100 km) versus the outer rainbands (r= 100-500 km) in a methodical manner throughout the manuscript. This should be done systematically for the entire analysis to better compare the simulation results with the observations and, in turn, establish more meaningful relationships between intensity fluctuations and lightning activity produced by the current model in the context of TCs. When the authors state "there was more lightning at X hour", I kept on wondering where and what was the inner core to outer band lightning ratio?

III. Emphasis should also be placed on defining proper lightning metrics (i.e., flash rates, flash density rates etc . . .) to establish more accurate comparisons with those from the WWLLN (pulse rates) (i.e., apples to apples comparison). Mentioning that WWLLN detects xx "lightning" is meaningless; same for LPI (in J/kg). I'd strongly advo-

cate converting all to flash rate or a metric that is more palpable and, thus, comparable to obs [e.g., pulse rate, flash origin density rates . . . etc].

IV. Many observational and modeling works on storm scale electrification have shown that graupel and updraft volume were the bulk quantities exhibiting the best correlations with total lightning and, as such, should be shown in this analysis for both the inner core and outer region/rainband to provide a more adequate diagnosis of the relationship(s) between lightning activity and intensity fluctuations (see e.g., Fierro an Mansell 2018). The authors have the model output data to do so.

V. The lightning scheme used is very basic as it is 100% diagnostic and does not take into account any fundamentals of lightning physics; e.g., 3D electric field solve, computation of polarization and noninductive charging rates, charge advection/sedimentation, lightning discharge processes etc . . .all of which were shown – in the context of a bulk discharge scheme - to be computationally efficient in the WRF framework (Fierro et al. 2013) and, thus, could easily be implemented in the Thompson scheme herein as contrarily indicated in the conclusion section. The authors should discuss this in more detail and work towards this goal for a more physically sound approach to forecast lightning in TCs or any other convective modes.

VI. From experience, I would argue that the McCaul diagnostic lightning scheme offers a more physically sound approach (better alternative) to lightning diagnosis than the LPI code via its inclination to e.g., give explicit consideration to stratiform and convective lightning (using graupel and ice fluxes). Furthermore, McCaul's scheme (implemented in WRF-ARW) has been battle tested in real time over several (∼8) years by many operational centers over the US and abroad for convective phenomena ranging from airmass thunderstorms to MCSs (including TCs). Thus, I fundamentally and respectfully differ that this is the "first" study investigating lightning in an operational model in general (but in HWR alone yes).

VII. What is the rationale for focusing on cases that are ∼10 years old for which no

total lightning data from the GLM are available? The 2017 year was very active with a near record number of major TCs (cat 3 or greater) many of which undergoing RI periods, ERCs, etc (Klotzbach, 2018). A shining example is Hurricane Maria, which total lightning activity with the GLM was studied in detail (Fierro et al. 2018) – including during its ERC - and contrasted to that of WWLLN's.

VIII. A. The distinction between intracloud and CG lightning is critical when studying lightning in any types of convective systems and should thus be carefully distinguished in the current study [in addition to outer region vs inner core in comment #II]: i.e., the model produces a surrogate for total lightning activity while WWLLN provides an estimate for total CG activity; especially over remote oceanic regions where DEs are low. Why would intracloud/total vs CG activity this be relevant for TCs ?: The aforementioned study on hurricane Maria, for example, underlined that intra-cloud to CG (or "Z") ratios could far exceed 10:1 in the inner core - which could change our perception on how lightning evolution relates to TC intensity. This is perhaps best exemplified by (and consistent with) the lightning jump algorithm for severe threat prediction (Schultz et al. 2011) almost entirely dependent on IC flash rates (See MacGorman and Nielsen 1991, MacGorman et al. 1989, Rutledge and Lang's seminal works etc) as CG flash rates alone only are indicative of the demise of an updraft (via reflectivity core descent). Boccippio et al. 2001 and Medici et al. 2017 found that in deep continental convection, IC flashes always outnumber CGs by a ratio sometime exceeding 10:1. Thus, it would make sense that when a VLF instrument such as the WWLLN detects a CG burst in the inner core, the updrafts are in their weakening stage as indicated in Fierro et al. (2011) for Hurricane Rita – and, thus the TC will undergo imminent weakening.

B. A more appropriate surrogate for evaluating the simulated total lightning activity from the model (either with LPI, McCaul or Fierro's explicit scheme) would be GLM lightning rates. The GLM instrument aboard GEOS-16/17 provides continuous day/night coverage of total lightning at ~90% detection efficiency (DE) over a large domain covering the Americas (Gurka et al. 2006; Goodman et al. 2012, 2013, Rudlosky et al.

2018). Similar space-borne technology to detect lightning have been developed by China (Feng-Yun-4, yang et al. 2016). Apart from their propensity to detect total lightning at a high DE, the chief advantage of this technology lies in its ability to retrieve lightning over remote oceanic regions where all TCs form and, eventually, intensify.

C. A and B and comment #II above illustrate that particular care should be given to total lightning in the inner core versus total lightning in the outer region / CG lightning in the inner core and outer region / CG lightning over the entire storm (r=0-500km) / total lightning over the entire storm. A proper study on TC lightning should make such distinctions very clear and evaluate these with available modeling and observational studies on TC lightning.

D. In the context of C, the authors should also provide statistics on which lightning behavior(s) listed above is (are) more systematically seen in the model during TC development, RI/intensification/weakening and why [using physical explanations based on eg internal dynamics or environmental factors]?

IX. The title is very generic and does not properly reflect the work done. I'd suggest something along the lines of: "Diagnostic forecasts of lightning activity within idealized and selected real tropical cyclone cases: preliminary results"

X. The results from Price et al. 2009 have been recently criticized by:

Whittaker I.C., E. Douma, C.J. Rodger, T.J.C.H. Marshall: A quantitative examination of lightning as a predictor of peak winds in tropical cyclones. J. Geophys. Res. Atmos., 120 (2015), pp. 3789-3801, 10.1002/2014JD022868

Which should be included/discussed wherever appropriate.

XI. The results section is almost completely devoid or references to previous modeling and observational works (e.g., TC Earl for which the lightning activity was studied in detail). Regarding the modelling (page 5, top) + no references whatsoever are provided for the various modules/parameterizations and vortex bogusing code used.

Because these issues are collectively substantial and would require thorough rewriting of the manuscript in many places, I opted not to dwell on editorial comments for the time being.

Figures:

There is no need to repeat in the body text what belongs to the figure captions. Please revise accordingly [eg page 9 bottom]. Figure 6 is very difficult to interpret due to the cluttering of contours.

Minor/Editorial comments:

Intro: Include a discussion on the effects of shear on TC lightning (modeling and observations).

Page 6: What is an inactive sounding ? initially at rest ?

Section 2.4, third line: consider revising (grammar).

Page 9: Please show how the secondary wind max "cuts off" the heat supply in HWRF. Invoking a reference is not sufficient. Either show it in your model data or delete the statement.

Respectfully, End of Review-

Please also note the supplement to this comment:
https://www.geosci-model-dev-discuss.net/gmd-2019-139/gmd-2019-139-RC1-supplement.pdf

---

## Referee Comment (RC2) · Anonymous Referee #2 · 27 Aug 2019

The Lightning Potential Index (LPI) parameterization is implemented in the HWRF model to investigate if lightning could help in reducing bias in intensity forecast. The use of lightning as a proxy for tropical cyclone intensification is of interest to the community. However, in its present form, this manuscript is not suitable for publication.

**General comments**

1. My first concern is: why did the authors decide to use a proxy for the lightning activity and not a complete electrical scheme (Mansell et al., 2002; Barthe et al., 2012; Fierro et al., 2013)? I guess that it is because of the high numerical cost of explicit electrical schemes. However, it should be stated clearly.

After making clear why a proxy for lightning activity is used, the choice of this proxy should be justified. Apart from the LPI, several microphysical and/or dynamical parameters have been used as proxies of the total flash rate: cloud-top height, maximum updraft speed, updraft volume, precipitation ice mass, graupel mass or volume, or even more complex parameters. Most of these parameters were discussed recently in Basarab et al. (2015), Lopez (2016) and Bovalo et al. (2019). A discussion about the choice of this parameter among all parameters available in the literature should be added.

2. One important limitation of this study is the choice of the case studies. Four different simulations are presented, but the analysis of each case study is too superficial. In my opinion, the authors should have treated only one case study: tropical cyclone Earl or a more recent case study. Concerning hurricane Igor, Figure 12 clearly shows that HWRF underestimates the analyzed intensity ($\sim$ 24 m s$^{-1}$ at 70 h) and does not capture the rapid intensification phase. Then, comparing the LPI with the intensity of the best-track does not make any sense. Focusing on one particular hurricane would allow to evaluate correctly the simulation and to conduct more useful diagnostics to support the conclusions.

Due to the relatively poor detection efficiency of the total flash rate with WWLLN and the availability of total lightning data from GLM, a recent hurricane of the Atlantic basin could have been chosen as a case study.

3. The presentation of the results is not well structured. For example, the model validation (track and intensity only) comes after the search for correlation between the lightning activity and the hurricane intensification (for Earl and Igor).

4. The results are not convincing.

- The authors claims that a "lightning maximum occurs prior to the intensity peak", but, as shown in Figures 1, 12 and 14, there are several maxima and minima of lightning activity before the maximum intensity.

- For all cases, it is not possible to check whether the cyclone has reach its maximum intensity at the end of the forecast time.

- The evolution of the idealized hurricane structure shown in Figure 3 is very weird. After 110 hours of simulation, there is no more rainband in the system which can explain why there is no LPI in the rainband (p 8, l 29-30)!

- An eyewall replacement cycle (ERC) is obtained when changing the planetary boundary layer scheme. However, the impact of the ERC on the intensity is not visible, and it is difficult to see an ERC on Figure 6.

5. The references are not always appropriate: see the specific comments. The introduction needs to be deeply modified.

For all those reasons, I think this manuscript is not suitable for publication. However, the topic is of great interest for the community, and the authors could resubmit this paper after a substantial work.

**Specific comments**

p 3, l 11-12: The formation of lightning happens in convective cells in general and not only in tropical cyclones.

p 3, l 13: The references (MacGorman and Rust, 1998; Rakov and Uman, 2003) are not about convection in tropical cyclones.

p 3, l 16-18: more recent references should be added like Jiang et al. (2013), Bovalo et al. (2014) and Zhang et al. (2015).

p 3, l 28-29: the original reference for the definition of rapid intensification is Kaplan et al. (2010).

p 3, l 26-34: more physical explanations about the two different theories are expected. For example, how an increase in lightning activity in the outer rainband can be linked to a rapid intensification phase?

p 4, l 16-18: "... the lightning potential is a diagnostic tool, it cannot change the model forecast of a tropical cyclone". Of course! This must be checked out during the developing phase of the model, but it is not necessary to discuss about this in the manuscript.

p 4, l 25: the HWRF model is an atmosphere-ocean coupled model, but there is nothing about the numerical set up of the ocean model in the manuscript.

p 4, l 28: The references (Tallapragada et al., 2014, 2015) are about WRF and not HWRF.

p 5, l 1-3: what is the difference between 18-km, 6-km and 2-km resolution, and grid spacing of 0.135°, 0.045° and 0.015°, respectively? Grid spacing and resolution should not be used interchangeably: https://doi.org/10.1175/1520-0477(2000)081%3C0579:CAA%3E2.3.CO;2

p 5, l 19-20: More suitable references for charge separation could be Saunders (2008) and references included.

p 6, l 3: The LPI parameterization was introduced in the Thompson microphysics scheme while the Ferrier-Aligo microphysics scheme was specifically designed for tropical cyclones (see p 5, l 6-7). This choice should be justified.

p 6, l 30-31: The term intra-cloud flashes or (cloud flashes) is generally used to refer to flashes that do not reach the ground (see MacGorman and Rust (1998)).

p 9, l 31-37 and p 10, l 1-4: there is no link with the discussion.

Since the results are not convincing, I do not go beyond the third section in my specific comments.

**Technical corrections**

The manuscript needs a careful proofreading.

The authors should rework on most of the figures:

- Figures 1 and 4: remove "solid" and "dash" in the titles of the Y-axis

- Figures 1, 2 and 4: make sur the aspect ratio of these figures is 1

- Figure 2: the unit of LPI is missing

- Figure 4: it is hard to distinguish between the solid and dash lines. A grey line should be used like in Figure 1.

- Figure 5: it's difficult to read the a-h numbering of the panels

- Figure 6: there are too many isocontours for radar reflectivity. Only the most important contours (to visualize the ERC) should be kept on this figure. There are two overlaid "130h", "136h", "142h", "148h" and "154h" on some panels.

- Figure 9: check the location of the titles on the panels

- Figures 11 and 14: remove "blue dashed line" in the title of the Y-axis. Add the unit of the flash rate from WWLLN in the caption.

- Figure 13 is not discussed in the manuscript. There are not dots every 6 hours as mentionned in the caption.

**References**

Barthe, C., Chong, M., Pinty, J.-P., Bovalo, C., and Escobar, J.: CELLS v1.0: updated and parallelized version of an electrical scheme to simulate multiple electrified clouds and flashes over large domains, Geosci. Model Dev., 5, 167–184, 2012.

Basarab, B. M., Rutledge, S. A., and Fuchs, B. R.: An improved lightning flash rate parameterization developed from Colorado DC3 thunderstorm data for use in cloud-resolving chemical transport models, J. Geophys. Res. Atmos., 120, 9481–9499, doi:10.1002/2015JD023470, 2015JD023470, 2015.

Bovalo, C., Barthe, C., Yu, N., and Bègue, N.: Lightning activity within tropical cyclones in the South West Indian Ocean, J. Geophys. Res. Atmos., 119, 8231–8244, doi:10.1002/2014JD021651, 2014.

Bovalo, C., Barthe, C., and Pinty, J.-P.: Examining relationships between cloud-resolving model parameters and total flash rates to generate lightning density maps, Quarterly Journal of the Royal Meteorological Society, 145, 1250–1266, doi:10.1002/qj.3494, https://rmets.onlinelibrary.wiley.com/doi/abs/10.1002/qj.3494, 2019.

Fierro, A. O., Mansell, E., MacGorman, D., and Ziegler, C.: The implementation of an explicit charging and discharge lightning scheme within the WRF-ARW model: Benchmark simulations of a continental squall line, a tropical cyclone and a winter storm., Mon. Wea. Rev., 141, 2390–2415, doi:10.1175/MWR-D-12-00278.1, 2013.

Jiang, H., Ramirez, E. M., and Cecil, D. J.: Convective and rainfall properties of tropical cyclone inner core and rainbands from 11 years of TRMM data., Mon. Wea. Rev., 141, 431–450, 2013.

Kaplan, J., DeMaria, M., and Knaff, J. A.: A revised tropical cyclone rapid intensification index for the Atlantic and Eastern North Pacific basins., Wea. Forecasting, 25, 220–241, 2010.

Lopez, P.: A Lightning Parameterization for the ECMWF Integrated Forecasting System, Mon. Wea. Rev., 144, 3057–3075, doi:10.1175/MWR-D-16-0026.1, 2016.

MacGorman, D. R. and Rust, W.: The electrical nature of storms., Oxford Univ. Press, 1998.

Mansell, E. R., MacGorman, D., Ziegler, C. L., and Straka, J. M.: Simulated three-dimensional branched lightning in a numerical thunderstorm model, J. Geophys. Res., 107, doi:10.1029/2000JD000244, 2002.

Saunders, C. P. R.: Charge separation mechanisms in clouds., Space Sci. Rev., 137, 335–353, 2008.

Zhang, W., Zhang, Y., Zheng, D., Wang, F., and Xu, L.: Relationship between lightning activity and tropical cyclone intensity over the northwest Pacific, Journal of Geophysical Research: Atmospheres, 120, 4072–4089, doi:10.1002/2014JD022334, https://agupubs.onlinelibrary.wiley.com/doi/abs/10.1002/2014JD022334, 2015.

---

## Referee Comment (RC3) · Anonymous Referee #3 · 16 Sep 2019

Summary: The authors implement the Lightning Parameterization Index (LPI) into the HWRF(V3.7a), and assess its performance in both idealized and real tropical cyclone case scenarios.

Recommendation: Accept, pending major revisions.

Main Comments:

The manuscript covers a topic of significance for the hurricane forecasting community. However, there are some major shortcomings in the main message. This is further stressed in the case selection, methodology, and choice of verification/validation data.

1. Not enough discussion or results aimed at the main stated purpose here: To reduce TC intensity forecast errors and bias. The results did not appear to show any additional

useful information for forecasters and model developers on the usefulness (or not) of having LPI implemented in the operational HWRF. This is in part due to a poor case selection, and an discussion that was not focused or concise enough around this point. For example, the authors go through a range of topics and cases, that give the reader a hard time to understand what exactly are they trying to accomplish. An ideal case with eyewall replacement cycle (ERC) is used, but both real cases shown here had ERCs (see NHC preliminary storm reports), and this is somehow ignored in the text. Why?

2. The case selection is puzzling. Initially, I thought this was going to be justified by use of data from the field campaigns of 2010 (e.g. NASA GRIP, NCAR PREDICT), in which a significant amount of in situ data were collected (especially on hurricane Earl). Having such old cases hinders the authors from the use of more robust lightning data sets that are currently available thanks to satellite sensors (e.g. GLM), which would be of great value to this study. There are plenty significant hurricane cases in the Atlantic that would make for a better selection. See, for example, hurricanes Irma, Jose, and Maria of 2017, or Michael in 2018.

3. The methodology is inconsistent and lacks depth. For example, in Fig. 1 the LPI is shown for the entire storm, instead of per storm region (e.g. inner core or eyewall, rainbands, periphery, etc.). This despite the discussion of lightning evolution in different regions (e.g. rainbands) during specific periods in the lifetime of the selected cases (e.g. rapid deepening vs. steady state). Azimuthal averaging was used for the ideal case, but not on the real cases. This should be addressed in order to have a more clear picture of the findings.

4. The verification/validation work is not robust at all. This is understandably related to the case selection (discussed above). More specifically, on hurricane Earl there was no mention of the ERC that strongly modulated the storm intensity/structure on August 30-31st, effectively ending the rapid intensification process, which is of focus in this study for its implications on lightning behavior. Evidently, the occurrence of the ERC meant major structural changes in Earl as well. None of this is properly

addressed here. No use of San Juan, Puerto Rico (TJUA) Doppler radar data is done to verify the HWRF reflectivity structure, nor the WWLLN lightning locations. For Igor, no land-based radar data is available, but microwave imagery would have been of use to validate the simulated storm structure. For the latter I suggest the NRL Monterrey TC Page archives, which do not require additional plotting. The main point here is: It is hard to credibly propose a set of conclusions without a more rigorous examination of the selected cases.

Specific comments:

Suggest adding a table listing each simulation and its main specifics for easier following.

Page 3, Line 8-9: Define "long/short time scales".

Page 3, Line 28 (and beyond): The use of kts is fine, but please include m/s as well, especially when those are the units used in the figures of this manuscript.

Page 4, Line 5-11: No need to repeat the abstract. Suggest removing this entire section.

Page 4, Line 29-33: Too many unnecessary details. Suggest making it more concise. Once sentence should be enough for the reader.

Page 5, Line 12-13: "In order to obtain the center of the tropical cyclone the model is run." This sentence needs to be fixed.

Page 6, Line 3-5: Why was LPI not implemented in the Ferrier-Aligo (operational), and on Thompson instead? The purpose of this selection ought to be discussed for clarification. Also, how do these two microphysics schemes compare? Use of existing literature is suggested.

Page 7, Line 5 (and beyond): Suggest reducing the long date format use in the manuscript to mm/dd/yyyy hhhhUTC (or Z).

Section 2.5: Many details not relevant to this study are included here. A much more concise description of both storms, focused on the simulation periods, would be a nice improvement. Also, citing NHC preliminary storm reports should be enough for the more thirsty readers.

Page 8, Line 17: "lightning peak occurs hours before the intensity peak". Please briefly explain the physical process(es) responsible for such behavior.

Page 8 Line 34-36: The writing here is confusing. Are you referring to the lack of direct ERC observations (not true), or of lightning in hurricanes undergoing ERCs? Either way, more recent cases prove otherwise.

Page 9 Line 20: Suggest expanding on this statement. A series of events lead to the expansion of the tangential wind field, which shifts the focus of low-level convergence to the secondary eyewall radii. In addition to Bell et al. (2011), suggest reading Houze et al. (2006), Terwey and Montgomery (2008), and Huang et al. (2012).

Page 10 Line 2 (and rest of manuscript): "Inner core" and "outer region" need to be defined quantitatively.

Page 10 Line 15-16: Not sure I agree with this statement. LPI appears to peak in advance of maximum winds, which may not be depicted as a negative correlation, but instead as a lagged (or delayed) response in the behavior of the wind speeds.

Page 10 Line 18-21: Suggest briefly describing storm structure here, including ERC and its role on intensity changes.

Page 11 Line 1: Why are August 30 at 0400 and 0800 UTC discussed, but not shown on figure? Suggest addressing this by either modifying figures, removing discussion on these times, or clarifying that it's "not shown".

Section 3.4: The discussion on hurricane Igor results seems too brief and superficial. Suggest either removing this case altogether, or conducting a more meaningful discussion on findings for this simulation.

Page 11 Line 19: A negative correlation of just -0.29 seems to contradict your conclusion on Page 10 Line 15-16. This is a weak correlation, at best.

Page 11 Line 25-29: Again, findings strongly contradicting your own statement on LPI vs maximum wind correlations.

Page 11 Line 35-36: The incompleteness of WWLLN data strengthens the point I made before; better (and more recent) cases need to be added to this study.

Page 12 Line 20-22: This statement was never properly addressed in this study.

Page 12 Line 29-30: This is a finding that is more consistent with the results presented herein. Suggest building on it across the text of this manuscript.

Page 13 Line 4: Not intending to beat a dead horse, but this is exactly one of the main issues with the current version of this manuscript.

Page 13 Line 7-10: This is a puzzling statement. How can a diagnostic tool dependent on the model itself be a warning for the own model's forecast error?

Page 13 Line 21-22: This has not been proved here, at all. Need to include radar observations and microwave imagery in order to make such statement.

Figures:

Fig 2. Why is the storm exhibiting a Southern Hemisphere signature (i.e. clockwise rotation), while showing counter clockwise signature (N.H. cyclone) on Fig. 6?

Fig. 6: More robust if showing an azimuthal average here as well. This will ensure a more complete picture of the storm structure and behavior.

References:

Houze RA Jr et al (2006) The hurricane rainband and intensity change experiment observations and modeling of Hurricanes Katrina, Ophelia, and Rita. Bull Am Meteorol Soc 87:1503–1521

[Figure]

Huang Y-H, Montgomery MT, Wu CC (2012) Concentric eyewall formation in Typhoon Sinlaku (2008). Part II: axisymmetric dynamical processes. J Atmos Sci 69:662–674

NHC Hurricane Earl tropical cyclone report: https://www.nhc.noaa.gov/data/tcr/AL072010_Earl.pdf

NHC Hurricane Igor tropical cyclone report: https://www.nhc.noaa.gov/data/tcr/AL112010_Igor.pdf

NRL Monterrey TC archives: https://www.nrlmry.navy.mil/tc_pages/tc_home.html

Terwey WD, Montgomery MT (2008) Secondary eyewall formation in two idealized, full-physics modeled hurricanes.   J Geophys Res 113:D12112. doi:10.1029/2007JD008897